# TCR-Engineered Lymphocytes Targeting NY-ESO-1: In Vitro Assessment of Cytotoxicity against Tumors

**DOI:** 10.3390/biomedicines11102805

**Published:** 2023-10-16

**Authors:** Alaa Alsalloum, Saleh Alrhmoun, Julia Shevchenko, Marina Fisher, Julia Philippova, Roman Perik-Zavodskii, Olga Perik-Zavodskaia, Julia Lopatnikova, Vasily Kurilin, Marina Volynets, Yasushi Akahori, Hiroshi Shiku, Alexander Silkov, Sergey Sennikov

**Affiliations:** 1Laboratory of Molecular Immunology, Federal State Budgetary Scientific Institution Research Institute of Fundamental and Clinical Immunology, Novosibirsk 630099, Russia; alaa.alsalloum19996@gmail.com (A.A.); saleh.alrhmoun1@gmail.com (S.A.); shevchenkoJA2023@yandex.ru (J.S.); msolshanova@gmail.com (M.F.); airyuka@mail.ru (J.P.); zavodskii.1448@gmail.com (R.P.-Z.); perik.zavodskaia@gmail.com (O.P.-Z.); lopatnikova18@yandex.ru (J.L.); vkurilin@niikim.ru (V.K.); mrsmarinavolynets@gmail.com (M.V.); silkov@niikim.ru (A.S.); 2Faculty of Natural Sciences, Novosibirsk State University, Novosibirsk 630090, Russia; 3Department of Personalized Cancer Immunotherapy, Graduate School of Medicine, Mie University, Tsu 514-8507, Japan; yakahori@med.mie-u.ac.jp; 4Department of Immunology, V. Zelman Institute for Medicine and Psychology, Novosibirsk State University, Novosibirsk 630090, Russia

**Keywords:** TCR, TCR-T cells, NY-ESO-1, T-cell activation, cellular cytotoxicity

## Abstract

Adoptive T-cell therapies tailored for the treatment of solid tumors encounter intricate challenges, necessitating the meticulous selection of specific target antigens and the engineering of highly specific T-cell receptors (TCRs). This study delves into the cytotoxicity and functional characteristics of in vitro-cultured T-lymphocytes, equipped with a TCR designed to precisely target the cancer-testis antigen NY-ESO-1. Flow cytometry analysis unveiled a notable increase in the population of cells expressing activation markers upon encountering the NY-ESO-1-positive tumor cell line, SK-Mel-37. Employing the NanoString platform, immune transcriptome profiling revealed the upregulation of genes enriched in Gene Ontology Biological Processes associated with the IFN-γ signaling pathway, regulation of T-cell activation, and proliferation. Furthermore, the modified T cells exhibited robust cytotoxicity in an antigen-dependent manner, as confirmed by the LDH assay results. Multiplex immunoassays, including LEGENDplex™, additionally demonstrated the elevated production of cytotoxicity-associated cytokines driven by granzymes and soluble Fas ligand (sFasL). Our findings underscore the specific targeting potential of engineered TCR T cells against NY-ESO-1-positive tumors. Further comprehensive in vivo investigations are essential to thoroughly validate these results and effectively harness the intrinsic potential of genetically engineered T cells for combating cancer.

## 1. Introduction

Advancements in human cell gene modification technologies have led to significant progress in the field of immunotherapy for oncological diseases. The ability to manipulate the structure and specificity of the T-cell receptor has enhanced the effectiveness of adoptive T-cell therapy approaches and significantly expanded their potentials [1,2]. Promising results have been achieved in the treatment of B-cell leukemias by using T cells with a chimeric antigen receptor (CAR) due to CAR’s high efficiency in recognizing surface antigens and initiating a potent specific immune response [3,4].

Nonetheless, the effectiveness of CAR-T cells for solid tumor therapy is limited due to the insufficient presence of surface molecules suitable for CAR targeting. Moreover, the CAR receptor’s mechanism of MHC-independent antigen recognition hinders the ability to target intracellular antigens, which are primarily found within solid tumors [5].

Within this context, the employment of genetic modification to induce the expression of a particular T-cell receptor (TCR) in T cells initially attuned to distinct antigenic determinants presents a notable advantage. This process facilitates the generation of a substantial quantity of T-lymphocytes, all characterized by a uniform TCR specificity for the selected intracellular tumor antigen, despite the heterogeneity of the T-cell population [6].

Currently, a variety of methodologies are accessible for delivering genetic constructs encoding antigen-specific TCRs. These encompass non-viral systems such as Sleeping Beauty and Piggy Bac, employing transposon-mediated mechanisms, as well as advanced methods like CRISPR-Cas9 and TALEN. Nonetheless, it is noteworthy to underscore that retroviral vectors harboring CAR or TCR sequences persist as the primary choice, prominently utilized in both preclinical investigations and clinical trials [7].

This paper describes the generation, along with the phenotypic and functional properties of lymphocytes with specialized TCR-T cells targeting the NY-ESO-1 antigen in the context of HLA-A*02:01.

NY-ESO-1 has been observed in various malignancies, including head and neck cancer, myeloma, metastatic melanoma, and breast cancer, and others [8,9,10,11,12]. NY-ESO-1 represents a typical cancer-testis antigen, prevalently manifested within immune-restricted niches such as the testes and placenta [13]. This distinct localization in normal tissues, coupled with its broad distribution across diverse tumor types, positions NY-ESO-1 as a valuable target with limited off-target toxicities and wide-ranging utility in various cancer types.

In our study, we utilized in vitro transduction with viral vectors to generate TCR-T cells. Subsequently, we conducted a comprehensive examination to assess the phenotype, activation state, and cytotoxic capability of these TCR-T cells upon encountering NY-ESO-1-positive tumor cell lines. The findings of this study hold promise for improving cancer treatments. This potential approach could be further investigated in vivo by using genetically modified lymphocytes, thereby opening new doors for therapeutic exploration.

## 2. Materials and Methods

### 2.1. Study Population and Interventions

Venous blood sampling from healthy adult donors (*n* = 6) was approved by the local ethics committee of the Research Institute of Fundamental and Clinical Immunology at Meeting No. 129 on 17 February 2021. Each donor signed an approved informed consent form. All donors were of Caucasian race with permanent residence in Western Siberia, and the average age of the healthy blood donors was 27.33 ± 3.98 years (mean ± standard error of the mean) (Table 1).

### 2.2. Isolation of Peripheral Blood Mononuclear Cells

We collected venous blood samples of up to 5 mL and preserved them in EDTA-containing tubes. Peripheral blood mononuclear cells (PBMCs) were isolated from the whole blood using the conventional Ficoll-Urografin (PanEco, Moscow, Russia) density gradient method. Briefly, the peripheral blood was diluted with an equal volume of RPMI-1640 medium (Biolot, St. Petersburg, Russia). Following dilution, the blood was gently overlaid on a Ficoll-Urografin solution (ρ= 1.077 g/L) and centrifuged at 400 g and room temperature for 40 min. After centrifugation, the mononuclear cells were collected from the opalescent layer formed at the phase boundary, covering the entire cross-section of the tube.

### 2.3. Retroviral Vector

We obtained a gamma-retroviral vector harboring a synthetic T-cell receptor (TCR) insert specifically engineered to recognize the peptide the p157–165 (SLLMWITQV) derived from the tumor antigen NY-ESO-1, from Prof. H. Shiku at Mie University Graduate School of Medicine, Japan (Figure 1).

### 2.4. Triggering T-Cell Proliferation

We induced the ex vivo expansion of peripheral blood mononuclear cells (PBMCs) by coating retronectin at a concentration of 25 mg/mL (Takara Bio, Kusatsu, Japan) and CD3 antibodies at a dose of 5 mg/mL (Biolegend, San Diego, CA, USA) onto the wells of 12-well plates (TPP, Trasadingen, Switzerland). The PBMCs were cultured at a concentration of 0.5 × 10^6^ cells/mL in 2 mL of GT-T551 medium (Takara Bio, Japan) supplemented with 300 U/mL IL-2 (Roncoleukin, Biotech, Moscow, Russia) and 0.6% human blood serum of group AB. Subsequently, the plates were incubated in a controlled environment with 5% CO_2_ at 37 °C. During the 2nd and 3rd days of cultivation, we performed regular medium exchanges, replacing half of the culture medium volume with a fresh aliquot of IL-2 to sustain cellular proliferation and metabolic activity.

### 2.5. Transduction of Anti-CD3 Primed PBMCs via Retroviral Vectors

To perform retroviral transduction, we thawed 1 mL of a retrovirus solution in a water bath at 37 °C. Then, we diluted it four times in PBS (Biolot, St. Petersburg, Russia) with 2% human serum albumin (Microgen, Tomsk, Russia), and 5% glucose-citrate buffer. We added the diluted retroviral solution to retronectin-coated wells of 24-well plates and centrifuged the plate with the retrovirus solution for 2 h at 32 °C at 2000× *g* (Jouan MR 23, Nantes, France). After the liquid content was removed, the wells were washed twice with PBS containing 2% albumin. Then, we added 1.5–2 × 10^5^ PBMCs primed with anti-CD3 antibodies to 1.5 mL of medium supplemented with IL-2 (300 U/mL), and the cell suspension plates were centrifuged at 1000× *g* for 10 min at 32 °C. The cell plates were placed in a humidified atmosphere with 5% CO2 at 37 °C. The next day, a second round of transduction was performed by transferring the cells into the wells of new 24-well plates with retroviral particles. The cells were then centrifuged at 1000× g for 10 min at 32 °C, which allowed for an increase in the efficiency of transduction of 10–15%. After that, the cells were incubated in a humid atmosphere at 37 °C containing CO2. Following 6–8 h, the cells were transferred to new 6-well plates (TPP, Switzerland) supplemented with an equivalent volume of GT-T551 culture medium and 300 U/mL IL-2. On Days 9–10 after initiating the protocol, an aliquot of cells was collected to assess transduction efficiency and determine the cell phenotype. On the 11th day from the start of the protocol, the transduced cells were co-cultured with tumor cell lines to evaluate their cytotoxic activity in vitro. As a control, we used cells primed with anti-CD3 and stimulated with IL-2, to which no retrovirus was added and no transduction was performed.

### 2.6. Evaluation of the Efficiency of Transduction

We employed MHC-biotin tetramers specific to the studied NY-ESO-1 TCR, conjugated with streptavidin-PE to assess the transduction efficiency of PBMCs. The cells were stained with these tetramers at a 1:100 dilution and incubated for 20 min in the dark at room temperature. The MHC tetramers were provided by Prof. H. Shiku (Mie University Graduate School of Medicine, Mie, Japan). After two washes, we added streptavidin-PE (Biolegend, San Diego, CA, USA), at a dilution of 1:600, along with monoclonal antibodies targeting surface markers (anti-CD3-AF700 (Biolegend, San Diego, CA, USA)) and Zombie Aqua vital dye (Biolegend, San Diego, CA, USA) to evaluate the culture’s viability. Staining was conducted for 20 min at room temperature in the dark. The stained cells were then washed with PBS containing 0.1% NaN_3_ and subjected to analysis using an Attune NxT flow cytometer (Thermo Fisher, Waltham, MA, USA). We gated cells from debris, singlets from the cells, alive cells from the singlets, CD3-positive cells from the alive cells, and MHC tetramer-positive cells from the CD3-positive cells (Figure 2). We observed a 22.13 ± 7.11% retroviral transduction of said T cells (mean ± standard deviation, *n* = 6), with a minimum value of 13.40% and a maximum of 33.5%.

### 2.7. Cell Lines

We utilized the human melanoma cell lines NW-Mel-38 and SK-Mel-37 as cell lines that express the NY-ESO-1 target antigen. For our experiments, we used the HCT-116 cell line as a negative control, which does not express the NY-ESO-1 target antigen. The tumor cell lines were cultivated in culture flasks with a planting concentration of 50–75 thousand cells/mL until they reached the log-phase of growth. RPMI-1640 medium supplemented with 10% FCS (Hyclone, Logan, UT, USA), 2 mM L-glutamine (Biolot, St. Petersburg, Russia), 5 × 10^−5^ mM mercaptoethanol (Sigma, Cibolo, TX, USA), 25 mM HEPES (Sigma, Cibolo, TX, USA), 80 µg/mL gentamicin (Krka, Novo Mesto, Slovenia), and 100 µg/mL ampicillin (Sintez, Kurgan, Russia) was used for cell culture. After detachment from the plastic surface using trypsin-versen solution (Biolot, St. Petersburg, Russia), the cells were seeded into the wells (4–5 × 10^4^ cells/well) of a 96-well flat-bottomed culture plate (TPP, Trasadingen, Switzerland) for 16–17 h, ensuring proper adhesion of the tumor cells to the plastic. The cell lines NW-Mel-38 and SK-Mel-37 were obtained from the collection of cell cultures at the Graduate School of Medicine, Mie University, under the supervision of Professor H. Shiku.

### 2.8. Phenotyping for Markers of Activation and Cytotoxicity by Flow Cytometry

We aimed to explore the antigen-dependent activation level and cytotoxic potential of the transduced T lymphocytes by co-culturing the studied T cells with tumor target cells. The adherent tumor cells (target) were initially cultured for 16–17 h, after which they were replaced with fresh culture medium, and T cells (effectors) were introduced to achieve an effector: target ratio of 5:1. We co-cultured the transduced T cells with the human tumor lines SK-MEL-37, NW-Mel-38, and HCT-116 for 4–5 h. We conducted an analysis of activation and cytotoxicity marker expression using flow cytometry, employing a range of monoclonal antibodies, including anti-CD107a-APC/Cyanine7 (cat#328630), anti-CD3-AF700 (cat #300324), anti-CD8-PeCy7 (cat #344712), anti-CD4-Bv570 (cat #300534), CD137(4-1BB)-BV711TM (cat #309832), anti-CD154 (CD40L)-PerCP/Cy5 (cat #310834), anti-CD69-AF647 (cat #310918), and anti-CD178(FasL)-BV421TM (cat #306412). All antibodies were sourced from Biolegend (USA) and used in strict accordance with the manufacturer’s instructions. For NY-ESO-1-specific cell labeling, we utilized anti-Fab as previously described. Following the staining process, the cells were washed with PBS containing 0.1% NaN_3_ and subsequently analyzed using an Attune NxT flow cytometer (Thermo Fisher, USA). We then gated cells from debris, singlets from the cells, alive cells from the singlets, and CD3-positive cells from the alive cells and exported them as .fsc files in the conventional gating software for the Attune NxT Flow cytometer. We then transformed .fcs files to .csv files using a custom Python 3 code via Jupyter Notebooks. We performed arcsinh-transformation with the automatically selected cofactors of the flow cytometry data contained in the.csv files to automatically split cells into negative and positive for all the markers with the R script (https://github.com/janinemelsen/Single-cell-analysis-flow-cytometry/blob/master/scripts/CSV_to_transformed_normalized_FCS_git.R, accessed on 16 July 2023) published by Melsen et al. [14]. We then performed simultaneous batch correction and data normalization by fdaNorm and exported the corrected .csv files as .fcs with the R script (https://github.com/janinemelsen/Single-cell-analysis-flow-cytometry/blob/master/scripts/clustering_dimensionalityreduction_pseudotime_git.R, accessed on 16 July 2023) originally published by Melsen et al. [14].

### 2.9. HSNE Dimensionality Reduction and Clustering

We normalized .fcs files into the Cytosplore app for macOS [15] and subjected them to an HSNE dimensionality reduction. We used CD4, CD8, CD40L, CD69, CD107a, CD137 (4-1BB), and FasL (CD178) for dimensionality reduction. We then exported the frequencies of the cells per cluster and visualized them in GraphPad Prism 9.4 as box plots.

### 2.10. Magnetic Separation of Transduced T Cells after Co-Culturing with Tumor Cells

To investigate the gene expression profile, we isolated NY-ESO-1-positive cells from the overall cell population following retroviral transduction. We accomplished this by adding an MHC-biotin tetramer to the cells on Day 11 of the experiment (10 µL of MHC-biotin tetramer per 10 × 10^6^ cells), followed by a 20-min incubation in a cold Versene solution supplemented with 0.5% bovine serum albumin. After two washes, we added magnetic beads conjugated with MojoSort™ Streptavidin Nanobeads (Biolegend, USA, cat #480016) to the cells at a rate of 10 µL of beads per 10 × 10^6^ cells. We then washed and sorted the cells using a MojoSort™ Magnet (Biolegend, USA, cat #480019). The sorted cells were allowed to rest for 16–17 h in a culture medium supplemented with IL-2 (300 U/mL). Next, we co-cultured these cells with adherent NY-ESO-1-positive tumor cells (SK-Mel-37) at an effector-to-target ratio of 5:1, maintaining the co-culture for 2 h to activate key genes that regulate the immune response. Following the co-cultivation with tumor cells, we separated the transduced T cells from the tumor cells using intensive pipetting in PBS. To eliminate any tumor cell contamination, we conducted positive magnetic sorting for the MojoSort™ Human CD45 Nanobeads (Biolegend, USA, cat #480029) marker. The viability of the magnetically sorted cells was then assessed using a Countess 3 Automated Cell Counter (Thermo Fisher Scientific, Waltham, MA, USA) and trypan blue staining, revealing a cell viability of over 92%.

### 2.11. Total RNA Extraction

We isolated total RNA from 300.000 to 600.00 cells with the Total RNA Purification Plus Kit (Norgen Biotek, Thorold, ON, Canada). We measured the concentration and quality of the total RNA in each sample on a Nanodrop 2000 spectrophotometer (Thermo Fisher Scientific, USA). We froze the total RNA at −80 °C until analysis.

### 2.12. Gene Expression Profiling by Nanostring

We performed gene expression profiling with the help of the Nanostring nCounter SPRINT Profiler analytical system using 100 ng of total RNA from each sample. We used the nCounter Human Immunology v2 panel to analyze the total RNA samples. The nCounter Human Immunology v2 panel consists of 579 immune and inflammation-associated genes, 15 housekeeping genes, and 8 negative and 6 positive controls. The samples (*n* = 3–6) were subjected to an overnight hybridization reaction at 65 °C, where 5–14 μL of total RNA was combined with 3 μL of nCounter Reporter probes, 0–7 μL of DEPC-treated water, 11 μL of hybridization buffer, and 5 μL of nCounter capture probes (total reaction volume = 33 μL). After the hybridization of the probes to targets of interest in the samples, the number of target molecules was determined on the nCounter digital analyzer. We performed normalization and QC in nSolver 4 using added synthetic positive controls and the 15 housekeeping genes included in the panel. We then performed background thresholding on the normalized data to remove non-expressing genes. The background level was determined as: mean of all NEG controls + 2SD of all NEG controls + mean of the POS_E controls.

### 2.13. Differential Gene Expression Testing

We performed differential gene expression using multiple *t*-tests (with Q < 0.005) in GraphPad Prism 9.4 for macOS. The volcano plot was created in GraphPad Prism 9.4. The GSEA was performed using GSEApy [16].

### 2.14. LDH Cytotoxicity Assay

After allowing the adherent tumor cells (target) to incubate for 16–17 h, we replaced the culture medium with X-VIVO 15 serum-free medium (Lonza) and then added T cells (effectors) to achieve an effector: target ratio of 5:1. Cytotoxicity was assessed by determining the lactate dehydrogenase (LDH) activity in the conditioned medium of transduced T cells and the human tumor lines SK-MEL-37, NW-Mel-38, and HCT-116 for a duration of for 6–8 h, using the CytoTox 96 Non-Radioactive Cytotoxicity Assay (G1780, Promega Corporation). LDH activity was measured through a 30-min coupled enzymatic assay, which converts the tetrazolium salt INT into red formazan. The absorbance of visible light was determined using a standard 96-well plate reader (Varioskan, Thermo Fisher Scientific), and the intensity of color formed was found to be directly proportional to the number of lysed cells. For statistical analysis of LDH cytotoxicity, we employed one-way ANOVA with Dunnett correction for multiple testing (with Q < 0.05) using GraphPad Prism 9.4.

### 2.15. Quantification of Cytokine Production

We co-cultured transduced cells with SK-Mel-37 tumor cells at an effector: target ratio of 5:1 for 48 h. Co-culture supernatants were collected and immediately frozen until the time of analysis. We employed the LEGENDplex™ Human CD8/NK Panel (13-plex) with a Filter Plate kit (Cat. Number: 740267, Lot Number: B330268, Biolegend, USA) to assess the cytokine content in the conditioned media. The analysis was performed following the manufacturer’s instructions. For each sample, 25 µL of conditioned medium was used. Subsequently, we compared the resulting cytokine profiles for transduced cells cultured with SK-Mel-37 tumor cells to those of non-transduced cells cultured with SK-Mel-37 tumor cells, aiming to identify any differences in cytokine production between the two experimental conditions. Statistical analysis was conducted using GraphPad Prism 10.0.0 software (GraphPad Software, San Diego, CA, USA). The Mann–Whitney test was used to compare between the two groups. Cytokines were categorized based on their respective concentration levels. The data are presented as the median and the interquartile range.

## 3. Results

### 3.1. Anti-NY-ESO-1-Construct-Transduced T Cells Exhibit Cytotoxicity Indicators

We meticulously analyzed flow cytometry data from T-lymphocyte cells, utilizing the HSNE dimensionality reduction technique. After co-cultivating with the NY-ESO-1-expressing SK-Mel-37 cell line, an in-depth analysis was conducted on the transduced T cells. The specific objective was to accurately categorize cells based on their distinct marker expression profiles. The implementation of cluster analysis revealed distinctly identifiable subgroups within the transduced T-cell population (Figure 3).

Most notably, the dominant subgroup was mainly composed of CD8^+^ FasL^+^ CD107a^+^ CD40L^−^ CD69^−^ 4-1BB^−^ lymphocytes, constituting approximately 67.50 ± 3.89% of the overall T-cell count. Moreover, around 17.63 ± 5.921% represented CD8^+^ CD107a^+^ CD40L^−^ CD69^−^ 4-1BB^−^ FasL^−^ cells, while roughly 12.50 ± 5.230% were characterized as CD8^+^ FasL^+^ CD69^+^ CD107a^+^ CD40L^−^ 4-1BB^−^ lymphocytes. Additionally, within the T-cell population, CD4^+^ FasL^+^ CD40L^+^ CD69^−^ CD107a^−^ 4-1BB^−^ cells were observed at a level of about 2.370 ± 0.4759% (mean ± standard error of the mean, *n* = 6) (Figure 4).

### 3.2. Transcriptional Profiling Unveils Immunological Signatures within Transduced T Cells

We performed a differential gene expression analysis on the immune transcriptome of transduced T cells, first prior to, and subsequently after, co-cultivation with the SK-Mel-37 cell line expressing NY-ESO-1. Our criteria for significance were q-values < 0.005 and log2 (Fold change) values > 0.847 or <−0.847 (Figure 5).

The transcriptomic analysis brought to light a group of genes with altered expression levels, notably featuring upregulated genes, including: AHR, *APP*, *ATG5*, *ATG7*, *BCL6*, *BST1*, *BST2*, *C1QBP*, *C1R*, *C1S*, *C3*, *CCL2*, *CCL20*, *CD164*, *CD22*, *CD274*, *CD276*, *CD59*, *CD81*, *CD83*, *CD9*, *CEBPB*, *CFI*, *CIITA*, *CSF1*, *CTNNB1*, *CXCL1*, *CXCL10*, *CXCL11*, *CXCL2*, *EDNRB*, *FCGR2A*, *FN1*, *GBP1*, *HLA-DMA*, *HLA-DMB*, *HLA-DPA1*, *HLA-DPB1*, *HLA-DQA1*, *HLA-DRA*, *HLA-DRB1*, *HLA-DRB3*, *ICAM1*, *IFIH1*, *IFNGR1*, *IKBKAP*, *IL13RA1*, *IL1A*, *IL1B*, *IL1RAP*, *IL6*, *IL6ST*, *IL8*, *IRAK2*, *IRF1*, *ITGA6*, *LIF*, *LTBR*, *NCAM1*, *NFKBIA*, *NT5E*, *PDCD1LG2*, *PLAUR*, *PRKCD*, *PSMB5*, *PSMB7*, *PTK2*, *RELB*, *SMAD5*, *SOCS3*, *SPP1*, *TFRC*, *TGFBI*, *THY1*, *TLR2*, *TNFRSF9*.

On the other hand, the downregulated genes constituted: *ADA*, *ARHGDIB*, *B2M*, *BAX*, *BCL2*, *CASP1*, *CASP8*, *CCL5*, *CCND3*, *CCR1*, *CCR2*, *CCR5*, *CCR7*, *CD2*, *CD244*, *CD247*, *CD27*, *CD28*, *CD3D*, *CD3E*, *CD4*, *CD45R0*, *CD45RA*, *CD45RB*, *CD48*, *CD5*, *CD53*, *CD6*, *CD7*, *CD70*, *CD80*, *CD8A*, *CD8B*, *CD96*, *CD99*, *CFH*, *CISH*, *CSF2RB*, *CTLA4-TM*, *CX3CR1*, *CXCR3*, *CXCR4*, *CXCR6*, *DPP4*, *ETS1*, *FKBP5*, *FOXP3*, *GBP5*, *GNLY*, *GPR183*, *GZMA*, *HAVCR2*, *HLA-A*, *HLA-B*, *ICAM2*, *ICAM3*, *ICOS*, *IFITM1*, *IFNAR2*, *IKBKB*, *IKBKE*, *IKZF1*, *IL10RA*, *IL12RB1*, *IL16*, *IL18R1*, *IL18RAP*, *IL21R*, *IL2RB*, *IL2RG*, *IL4R*, *IRAK4*, *IRF4*, *ITGA4*, *ITGAL*, *ITGAM*, *ITGB2*, *JAK1*, *JAK2*, *JAK3*, *KLRB1*, *KLRC1*, *KLRC2*, *KLRC3*, *KLRC4*, *KLRK1*, *LAIR1*, *LCK*, *LCP2*, *LEF1*, *MALT1*, *MAP4K1*, *MAP4K2*, *MAPK1*, *MAPKAPK2*, *MBP*, *MUC1*, *MYD88*, *NCF4*, *NFATC2*, *NFATC3*, *PDGFRB*, *PECAM1*, *PRDM1*, *PRF1*, *PSMB10*, *PSMB8*, *PSMB9*, *PTGER4*, *PTPN22*, *PTPN6*, *PYCARD*, *RARRES3*, *S1PR1*, *SELL*, *SELPLG*, *SH2D1A*, *SIGIRR*, *SLAMF1*, *SLAMF7*, *SOCS1*, *STAT4*, *STAT5A*, *STAT5B*, *TAGAP*, *TBX21*, *TGFB1*, *TGFBR2*, *TLR1*, *TMEM173*, *TNFRSF11A*, *TNFRSF14*, *TNFRSF1B*, *TNFSF10*, *TNFSF12*, *TNFSF4*, *TNFSF8*, *TP53*, *TRAF5*, *TYK2*, *UBE2L3*, *ZAP70*.

Following this, we executed gene set enrichment analysis for the upregulated genes, specifically exploring gene ontology biological process terms (Figure 6, Table 2).

### 3.3. Anti-NY-ESO-1-Construct-Transduced T Cells Induce Antigen-Specific Cytotoxicity

We embarked on an inquiry into the antigen-driven activation and cytotoxic potential of genetically modified T lymphocytes. These modified T cells, serving as effectors, were co-cultured with the human tumor cell lines SK-Mel-37, NW-Mel-38, and HCT-116 for a duration of 6 to 8 h, while maintaining an equilibrated effector-to-target ratio of 5:1. Our comprehensive analysis unveiled a significant enhancement in the cytotoxic response specifically directed towards cell lines expressing the target antigen, when compared to the NY-ESO-1-negative line HCT-116. Intriguingly, no significant variations in cytotoxicity were found between the NY-ESO-1-expressing cell lines NW-MEL-38 and SK-MEL-37 (Figure 7).

### 3.4. Anti-NY-ESO-1 Construct-Transduced T Cells Release Cytotoxicity-Related Cytokines upon Encountering Tumor Cells

We carried out an extensive study on cytokine release in the conditioned medium from two separate groups: transduced cells and non-transduced cells, both cultured alongside SK-Mel-37 tumor cells. Our analysis revealed significant fluctuations in the secretion levels of crucial cytotoxicity-associated cytokines like granzyme, perforin, interferon, tumor necrosis factor (TNF), soluble Fas ligand (sFasL), and interleukin 17, 6 (Figure 8). Remarkably, the transduced cells manifested significantly augmented cytokine secretion levels in response to the SK-Mel-37 tumor cells, distinct from the non-transduced cells. 

## 4. Discussion

Previous efforts to target NY-ESO-1 through various immunotherapeutic approaches, such as protein and peptide vaccines, and adoptive T-cell therapy have encountered challenges within the scientific community [17,18,19]. It has become increasingly evident that addressing these limitations is essential, fueling the search for improved methodologies and novel avenues in cancer immunotherapy [20,21]. Therefore, the pursuit of anti-NY-ESO-1 therapy continues to be a topic of significant interest.

In the scope of this research paper, we directed our attention towards a meticulous analysis of flow cytometry data, employing the HSNE dimensionality reduction technique to unravel the elaborate cross-talk of T cells with the NY-ESO-1-expressing SK-Mel-37 cell line. The findings revealed unique subsets of T cells characterized by their distinctive marker expression patterns. The dominant subgroup consisting primarily of CD8^+^ FasL^+^ CD107a^+^ CD40L^−^ CD69^−^ 4-1BB^−^ lymphocytes, representing the majority of the T-cell count, suggests a robust cytotoxic response. The expression of CD107a, also known as LAMP-1, is linked with degranulation and cytotoxic granule exocytosis, indicative of cellular cytotoxicity [22,23]. Moreover, the presence of FasL indicates the capacity to induce apoptosis in target cells, further emphasizing their cytotoxic potential [24]. The CD8^+^ CD107a^+^ CD40L^−^ CD69^−^ 4-1BB^−^ FasL^−^ subset, albeit smaller in size, still signifies cytotoxicity, aligning with the concept that CD107a expression corresponds to cytotoxic granule exocytosis. Meanwhile, the CD8^+^ FasL^+^ CD69^+^ CD107a^+^ CD40L^−^ 4-1BB^−^ lymphocytes subset might denote a more activated state, given that CD69 serves as an early activation marker [25]. The presence of CD4^+^ FasL^+^ CD40L^+^ CD69^−^ CD107a^−^ 4-1BB^−^ cells, albeit in a smaller proportion, implies the involvement of helper T cells capable of supporting CD8+ cytotoxic T cells [26,27]. The expression of CD40L (CD154) on CD4+ T cells aligns with well-established scientific literature, emphasizing the sustained presence of CD40L on activated CD4+ T cells [28]. These distinct subsets highlight the intricate nature of the immune response triggered by the NY-ESO-1 antigen. The diverse composition of cytotoxic subsets signifies a coordinated endeavor to efficiently counter target cells.

Upon T-cell activation, distinct gene expression programs are initiated in a microenvironment-dependent manner, promoting cellular energy generation, biosynthesis, progression through the cell cycle, and eventual cell differentiation [29]. Our comprehensive analysis of differential gene expression in transduced T cells’ immune transcriptome before and after co-cultivation with NY-ESO-1-expressing SK-Mel-37 cells revealed evidence of immune function activation. Notably, upregulated genes were associated with crucial biological processes, including the IFN-γ signaling pathway [30] and T-cell proliferation and activation regulation. For instance, the upregulation of the lymphotoxin beta receptor (*LTBR*) gene aligns with previous studies, where increased LTBR expression enhanced T-cell functionality and antigen-specific responses [31]. *C1QBP*, another upregulated gene, plays a pivotal role in mitochondrial oxidative phosphorylation (OXPHOS) [32] and CD8+ T-cell differentiation through metabolic-epigenetic pathways [33]. This complexity underscores the need for further investigations into metabolic pathways governing CD8+ T cells.

Distinct subsets of T cells can be accurately characterized by their genetic profiles, which act as specific markers for each cellular subtype. The notable reduction in the expression of pivotal genes, namely *CCR7*, *CXCR4*, and *CD45RA*, *SELL*(L-selectin), *CD27*, and *CD28* is of paramount significance within our study. These genes, commonly linked to naive T cells [34,35,36,37]. This change in gene expression aligns with established insights into T-cell differentiation pathways [38] and provides strong evidence supporting the shift of T cells from a naive state to an effector state. Our findings are in line with prior studies that have highlighted the pivotal role of CD62L shedding as a critical regulatory mechanism governing T-cell responses to tumor cells. Upon encountering tumor lines, antigen-reactive T cells exhibit a rapid loss of CD62L expression, concomitant with the acquisition of CD107a (Figure 3) [39]. This dynamic shift signifies a transformation of T cells towards an activated, cytotoxic state, a phenomenon of paramount importance in the context of anti-tumor immune responses. Furthermore, a downregulation was noted in critical regulatory T-cell (Treg) function genes, including *FOXP3* and *TGFB1* [40,41]. In alignment with these genetic findings, our comprehensive analysis of various cytokine proteins indicated the absence of significant secretion levels of the anti-inflammatory cytokines IL-10 and IL-4 (Figure 8) [42,43], suggesting a potential decline in immunosuppressive Tregs within the T-cell population.

Furthermore, our investigation revealed the heightened transcriptional activity of genes that encode chemokines. These genes perform well-known functions in guiding T-cell trafficking, facilitating infiltration into tumor microenvironments, regulating migratory patterns, and coordinating complex cellular interactions [44].

Additionally, we observed a substantial increase in *FCGR2A* gene expression, which encodes CD32 receptors. CD32 receptors have a critical function in linking innate and adaptive immunity by attaching to the Fc portion of IgG. Our research indicates that transduced T cells can modulate CD32 receptor expression in line with earlier studies [45,46].

The transcriptome analysis also demonstrated a significant decrease in the expression of *ZAP70*, *CD247 (CD3-zeta)*, *CD4*, *CD8A*, *CD8B*, and *LCK* genes in transduced T cells co-cultured with SK-MEL-37 cells for two hours. This downregulation could regulate T cell activation through two mechanisms: termination of signal transduction and moderation of antigen-mediated T-cell activation. Earlier research studies [47] indicated that the rapid degradation of ZAP70 controls TCR interactions and sustains constant signaling even in the presence of excess antigen. In addition, another study highlighted the intricate nature of gene regulation during T-cell activation. For instance, activation signals, such as anti-CD3 + PMA, led to a transient decrease in TCR, CD4, and CD8 mRNA levels within 4 h post-activation, followed by their gradual re-expression [48]. These observations underscore the dynamic and complex nature of immune regulation, underscoring the need for further research to elucidate the precise molecular mechanisms of these gene expression changes over time.

The intricate determination of the optimal time for quantifying cytokine gene expression upon activation arises from the interplay between the heterogeneity of the peripheral T-cell repertoire and the specific applied stimulus [49,50]. Conventional studies recommend assessing cytokine gene expression 4 to 8 h after stimulation [51]. However, our research provides a new perspective. Initially, we performed a two-hour co-culture of T cells to uncover the initial immune responses in transduced T cells. Moving forward from this phase, we examined functional cytotoxicity via a multiplex assay during a 48 h co-culture, consistent with the timing documented for the granzyme B gene in previous investigations [51]. We hypothesized that this time frame would be sufficient to facilitate the expression of genes related to cytotoxicity at the protein level. The analysis of cytokine release patterns revealed an increase in the production levels of granzymes, perforins, and soluble Fas ligand (sFasL). Furthermore, a significant increase was observed in the levels of pro-inflammatory cytokines, including TNF, IFNG, IL17, and IL6, when compared to non-transduced T lymphocytes. These findings correspond with the established mechanism of T-cell-mediated cytotoxicity [52]. This expanded temporal resolution could shed light on the evolving nature of the response, potentially uncovering hidden regulatory mechanisms or cascades of events that contribute to the observed cytotoxicity. It is worth considering future experiments that incorporate additional time points.

In addition, the obtained NY-ESO-1-specific TCR-T cells demonstrated cytotoxic activity against tumor cell lines expressing the target antigen, indicating the antigen-specific nature of the cell’s activation.

Taken together, these results underscore the effectiveness of our approach and offer valuable contributions to the progress of modified T-cell-based therapies for cancer treatment. These insights provide important directions for further research and potential applications in clinical immunology.

## 5. Conclusions

During the course of this investigation, we conducted a comprehensive analysis of genetically modified T cells’ responses towards NY-ESO-1-expressing SK-Mel-37 cells, revealing distinct T-cell subsets characterized by robust cytotoxicity and activation capabilities. The application of advanced methodologies facilitated the exploration of intricate immune dynamics. Gene expression profiling revealed elevated immune activation and intricate metabolic pathways, implying a nuanced regulatory network. These findings were further substantiated through the examination of cytokine release patterns, which demonstrated a significant production of pivotal cytotoxic cytokines, inducing an antitumor response in an antigen-dependent manner. The obtained results powerfully underscore the inherent capacity of these genetically modified T cells to elicit antitumor reactivity, indicating the expediency of pursuing in vivo investigations to further explore and capitalize on the potential of these cell types in advancing effective approaches for cancer immunotherapy.

## Figures and Tables

**Figure 1 biomedicines-11-02805-f001:**
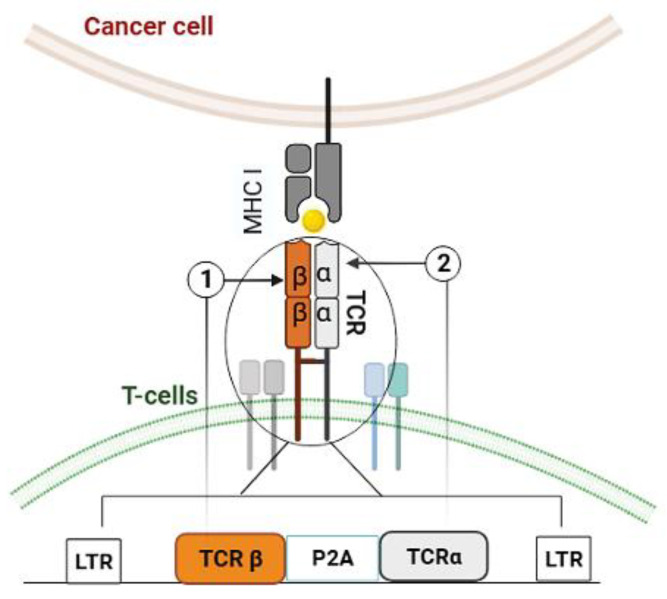
The scheme of the gamma retroviral vector (MMLV) encoding a T-cell receptor specific to the NY-ESO-1 epitope. The vector incorporates key components, including long terminal repeat (LTR), TCRβ (T cell receptor chain beta) ①, TCRα (T cell receptor chain alpha) ②, and P2A (self-cleaving peptide sequence).

**Figure 2 biomedicines-11-02805-f002:**
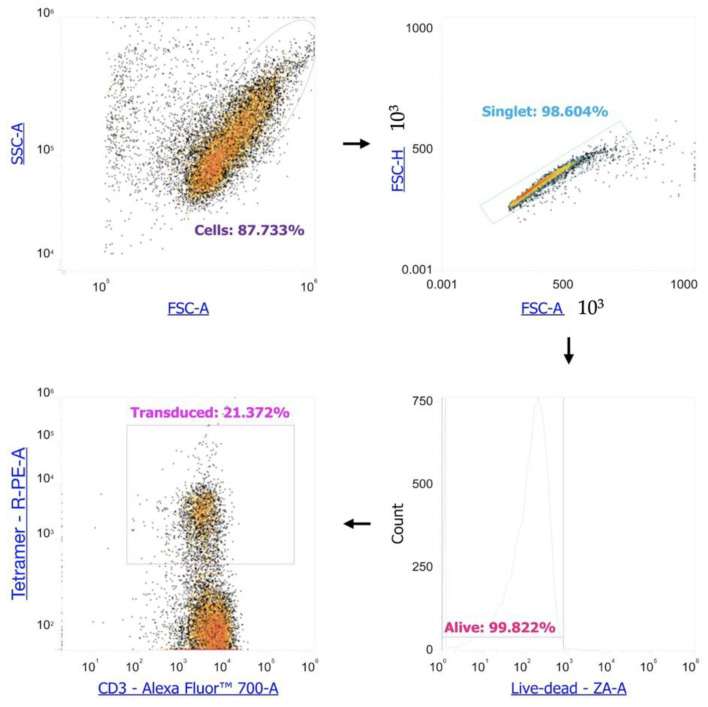
Retroviral transduction estimation of the anti-CD3 primed PBMCs by flow cytometry.

**Figure 3 biomedicines-11-02805-f003:**
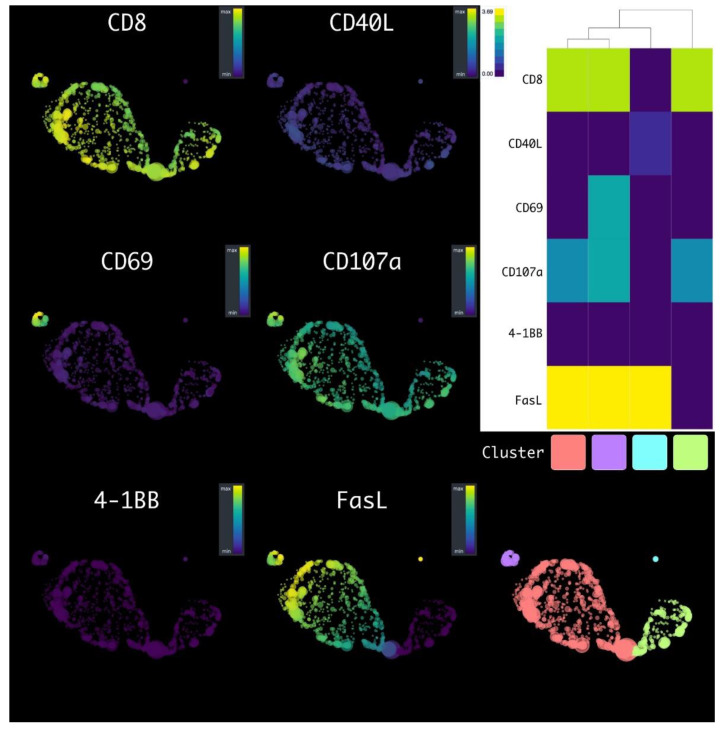
HSNE plots of the transduced T cells, after the co-cultivation with tumor cell lines, with a flow cytometry data markers’ overlay: clusters are color-labeled in accordance with the heatmap.

**Figure 4 biomedicines-11-02805-f004:**
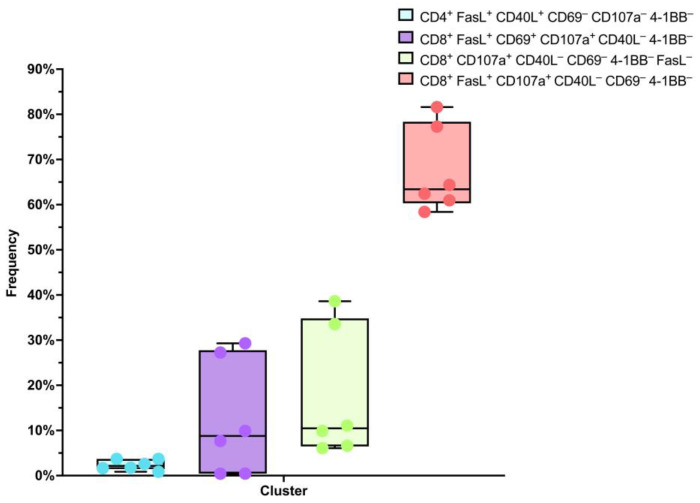
Box plot analysis of transduced T cell population clusters following NY-ESO-1-expressing SK-MEL-37 cell line cultivation.

**Figure 5 biomedicines-11-02805-f005:**
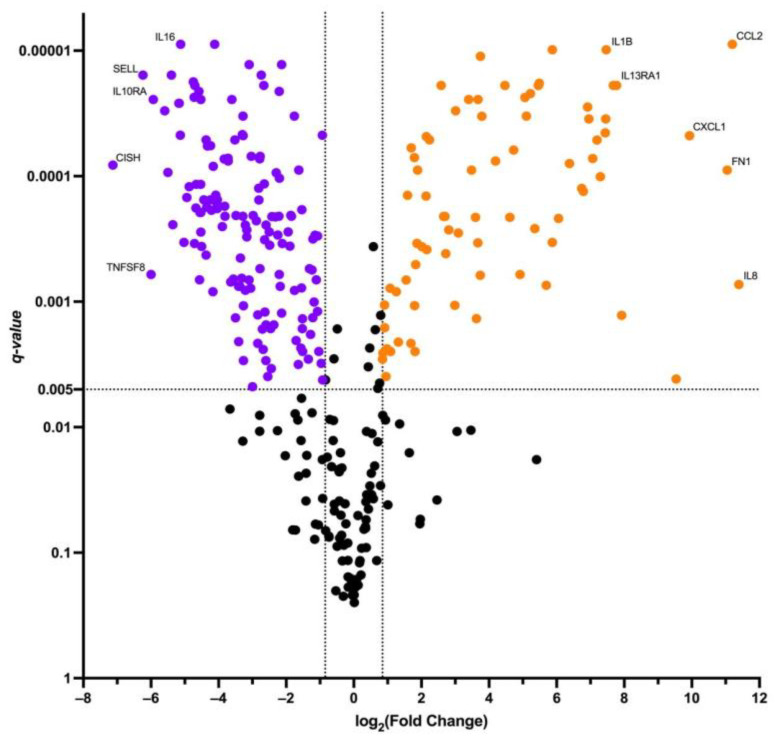
Volcano plot of differentially expressed genes. Orange dots depict upregulated genes, purple dots depict downregulated genes. Black dots represent non-significantly differentially expressed genes.

**Figure 6 biomedicines-11-02805-f006:**
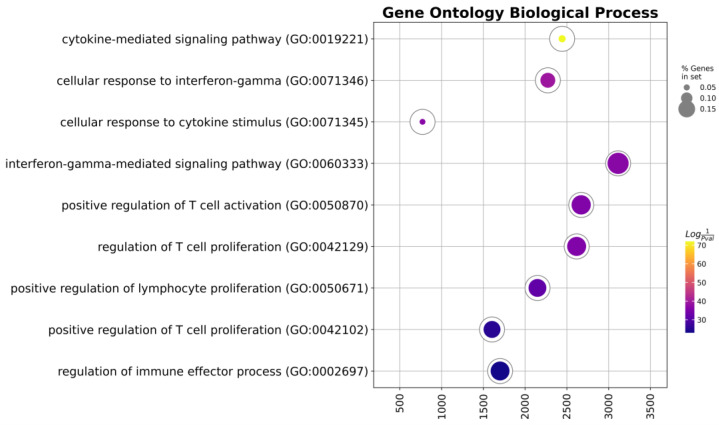
Ring plot of the gene set enrichment analysis of the upregulated genes.

**Figure 7 biomedicines-11-02805-f007:**
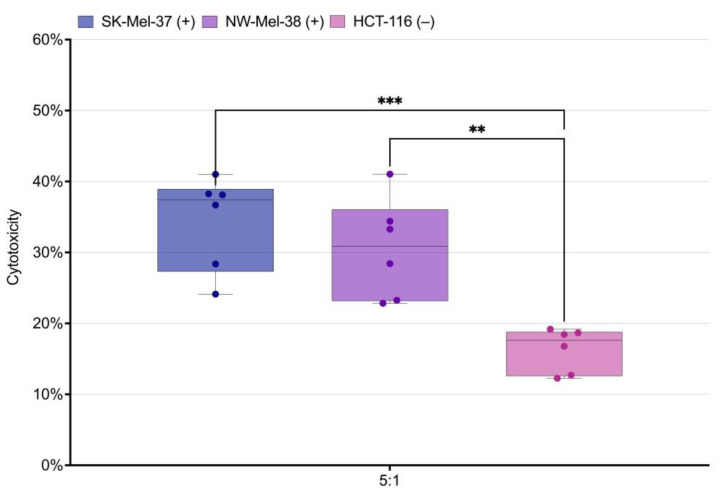
Analysis of LDH-measured cytotoxicity in the SK-Mel-37, NW-Mel-38, and HCT-116 cell lines, *** stands for *q*-value < 0.0005, ** stands for *q*-value < 0.001, (+) stands for the presence of the NY-ESO-1 expression, (−) stands for the absence of the NY-ESO-1 expression.

**Figure 8 biomedicines-11-02805-f008:**
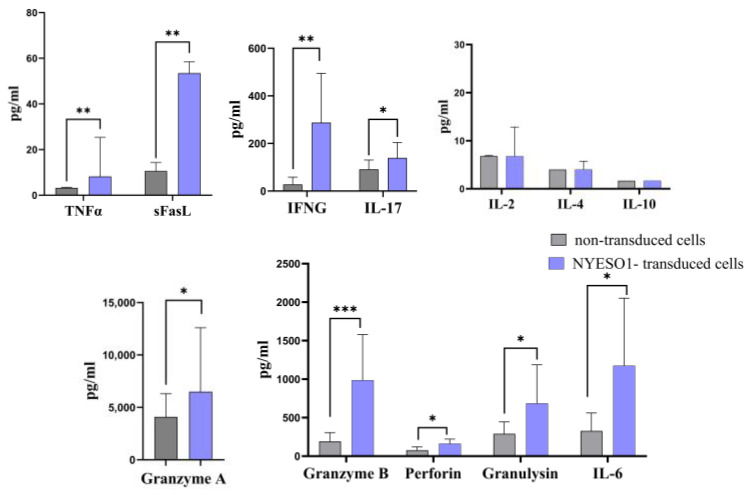
Bar plot of the cytokines secreted by transduced T cells following co-cultivation with tumor cell lines expressing the target antigen (SK-Mel-37). Each bar’s height represents the median levels of cytokines, and the error bars indicate the interquartile range. Significance levels are indicated by asterisks, with *** denoting a *p*-value < 0.001, ** representing a *p*-value < 0.01, and * indicating a *p*-value < 0.05.

**Table 1 biomedicines-11-02805-t001:** Demographic characteristics of donors.

Donor ID	Age (Years)	Gender	Previous Treatments	Race
D001	33	Male	No	Caucasian
D002	24	Male	No	Caucasian
D003	28	Male	No	Caucasian
D004	31	Male	No	Caucasian
D005	24	Male	No	Caucasian
D006	24	Male	No	Caucasian

**Table 2 biomedicines-11-02805-t002:** Gene set enrichment analysis of the upregulated genes.

Term	Overlap	*q*-Value	Combined Score	Genes
Cytokine-mediated signaling pathway	37/621	0.0	2443.644	*CIITA*, *CSF1*, *CXCL1*, *IL1RAP*, *CXCL2*, *ICAM1*, *SOCS3*, *PSMB7*, *PSMB5*, *IRAK2*, *CCL2*, *NCAM1*, *GBP1*, *HLA-DQA1*, *HLA-DPA1*, *IL13RA1*, *CCL20*, *IFNGR1*, *TNFRSF9*, *PRKCD*, *LIF*, *FN1*, *NFKBIA*, *BST2*, *IL1A*, *CXCL10*, *CXCL11*, *IL6*, *BCL6*, *IRF1*, *IL1B*, *HLA-DPB1*, *HLA-DRA*, *LTBR*, *IL6ST*, *HLA-DRB3*, *HLA-DRB1*
Cellular response to interferon-gamma	16/121	0.0	2272.529	*CIITA*, *CCL20*, *IFNGR1*, *PRKCD*, *ICAM1*, *IRF1*, *HLA-DPB1*, *HLA-DRA*, *CCL2*, *NCAM1*, *HLA-DRB3*, *GBP1*, *HLA-DQA1*, *HLA-DRB1*, *TLR2*, *HLA-DPA1*
Cellular response to cytokine stimulus	23/482	0.0	772.551	*CSF1*, *CCL20*, *IFNGR1*, *LIF*, *FN1*, *CXCL1*, *IL1RAP*, *CXCL2*, *ICAM1*, *IL1A*, *CXCL10*, *SOCS3*, *IL6*, *BCL6*, *IRAK2*, *IRF1*, *IL1B*, *CCL2*, *IL6ST*, *GBP1*, *TLR2*, *HLA-DPA1*, *IL13RA1*
Interferon-gamma-mediated signaling pathway	13/68	0.0	3113.543	*CIITA*, *IFNGR1*, *PRKCD*, *ICAM1*, *IRF1*, *HLA-DPB1*, *HLA-DRA*, *NCAM1*, *HLA-DRB3*, *GBP1*, *HLA-DQA1*, *HLA-DRB1*, *HLA-DPA1*
Positive regulation of T cell activation	13/75	0.0	2671.611	*CD274*, *TFRC*, *CD81*, *PDCD1LG2*, *THY1*, *IL6*, *HLA-DMB*, *IL1B*, *HLA-DPB1*, *CCL2*, *IL6ST*, *HLA-DPA1*, *CD276*
Regulation of T cell proliferation	13/76	0.0	2617.25	*CD274*, *CEBPB*, *TFRC*, *PDCD1LG2*, *IL6*, *HLA-DMB*, *IL1B*, *HLA-DPB1*, *IL6ST*, *HLA-DRB1*, *HLA-DPA1*, *IDO1*, *CD276*
Positive regulation of lymphocyte proliferation	12/75	0.0	2148.229	*CD274*, *BST1*, *IL6*, *HLA-DMB*, *TFRC*, *CD81*, *IL1B*, *HLA-DPB1*, *PDCD1LG2*, *IL6ST*, *HLA-DPA1*, *CD276*
Positive regulation of T cell proliferation	10/66	0.0	1604.452	*CD274*, *IL6*, *HLA-DMB*, *TFRC*, *IL1B*, *HLA-DPB1*, *PDCD1LG2*, *IL6ST*, *HLA-DPA1*, *CD276*
Regulation of immune effector process	9/53	0.0	1700.874	*C3*, *C1S*, *CD81*, *C1R*, *C1QBP*, *CFI*, *HLA-DRA*, *CD59*, *HLA-DRB1*

Note: *q*-value < 0.000001.

## Data Availability

The datasets generated and analyzed during the current research are accessible from the corresponding author upon an email request.

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
