# Peer review of "TCR-Engineered Lymphocytes Targeting NY-ESO-1: In Vitro Assessment of Cytotoxicity against Tumors"

_biomedicines, 2023, doi:10.3390/biomedicines11102805_

Round 1

Reviewer 1 Report

It is an interesting manuscript showing the effect of  TCR designed to  target the  cancer-testis antigen NY-ESO1 expressing SK-Mel-37 cells.  Using Nanostring, the authors showed that elevated immune activation and  intricate metabolic pathways, implying a nuanced regulatory network.

The authors concluded that genetically modified T-cells can elicit antitumor reactivity indicating potential anti-cancer activity.

Major points

1-  In Fig 8: what is the potential of using these cytokines in Figure 8? why these cytokines excatly? 

2- data of LDH assays is not presnted. please show them

3- Did the authors confirm the Nanostring data using qRT-PCR.

4- Demgraphic data of the 6 donors should be presented.

Moderate language editing

Author Response

Dear reviewer 1, We would like to thank you for taking the necessary time and effort to revise the manuscript. We sincerely appreciate all the valuable comments and suggestions, which helped us improve the quality of the work. All of the recommendations have been addressed in the manuscript. Revised parts are marked up using the "Track Changes" function.

Point 1: In Fig 8: what is the potential of using these cytokines in Figure 8? why these cytokines excatly?

Response 1:

You're absolutely right; this question, this question is highly significant for our research.

In our study, we used the LEGENDplex™ Human CD8/NK Panel (13-plex), which encompasses 13 key cytokines: IL-2, IL-4, IL-6, IL-10, IL-17A, IFN-γ, TNF-α, soluble Fas, soluble FasL, Granzyme A, Granzyme B, Perforin, and Granulysin. These cytokines are vital in immune regulation, cytotoxicity, and overall immune responsiveness, aligning closely with the objectives of our research, especially within the context of the highly prominent CD8+ lymphocyte subgroup.

 Our selection of the LEGENDplex™ Human CD8/NK Panel (13-plex) was guided by our cluster analysis, which highlighted a dominant subgroup primarily comprised of CD8+ lymphocytes within the transduced T-cell population. This panel played a crucial role in our research as it allowed us to precisely measure and quantify cytokine levels secreted by immune cells, with a particular focus on the highly prominent CD8+ lymphocyte subgroup. Importantly, these cytokines are directly linked to the significant findings presented in Figure 8, underlining their potential impact in our study.

Point 2: data of LDH assays is not presnted. please show them.

Response 2:

Your observation regarding the missing citation for the data in Figure 7 is much appreciated. We've resolved this issue by including the necessary citation. Furthermore, we've improved the figure's title for better clarity. You can track these changes using the 'Track Changes' function. Your feedback is a valuable contribution to enhancing the precision of our work, and we thank you for your thorough review.

Point 3:  Did the authors confirm the Nanostring data using qRT-PCR?

Response 3:

We appreciate your inquiry regarding the validation of our Nanostring data with qRT-PCR. In our study, we chose not to conduct qRT-PCR validation due to the well-documented reliability of Nanostring technology. There is substantial scientific evidence supporting the strong correlation between Nanostring and qRT-PCR results, as evidenced in a relevant study (https://bmcbiotechnol.biomedcentral.com/articles/10.1186/1472-6750-11-46).

Moreover,our data analysis was conducted rigorously, incorporating a strict cut-off criterion (mean of positive control E + mean of the Negative Controls + 2*standard deviations of the Negative Controls) to ensure data reliability. While we acknowledge the value of validation in research, the combination of Nanostring's established accuracy and our meticulous data analysis using stringent criteria provides us with a high degree of confidence in the precision of our results. Therefore, we believe these factors amply justify our decision not to pursue additional validation through qRT-PCR.

Point 4: Demgraphic data of the 6 donors should be presented.

Response 4:  We're truly grateful for your valuable input regarding the inclusion of demographic data. Our study primarily focuses on generating genetically modified cytotoxic T-cells, and we deliberately chose healthy donors to ensure T-cell quality. We believe labeling donors as 'healthy' and providing their average age (27.33 ± 3,98 years) is informative while respecting privacy. While we initially omitted the donors' gender and race, we acknowledge the significance of this information. In the revised manuscript, we will include that all donors were male, and all donors are from the Caucasian race with permanent residence in Western Siberia, as specified in Table 1. Your input is highly valued, and we will incorporate these details accordingly.

   Table 1. Demographic Characteristics of Donors

Donor ID

Age (years)

Gender

Previous Treatments

Race

D001

33

Male

No

Caucasian

D002

24

Male

No

Caucasian

D003

28

Male

No

Caucasian

D004

31

Male

No

Caucasian

D005

24

Male

No

Caucasian

D006

24

Male

No

Caucasian

Reviewer 2 Report

Authors performed a study including cytotoxicity and functional characteristics of in vitro-cultured T-lymphocytes, equipped with a TCR designed to precisely target the cancer-testis antigen NY-ESO1. Flow cytometry analysis unveiled a notable increase in the population of cells expressing activation markers upon encountering the NY-ESO-1-positive tumor cell line, SK-Mel-37. Employing the NanoString platform, immune transcriptome profiling revealed the upregulation of genes enriched in Gene Ontology Biological Processes associated with the IFN-γ signaling pathway, regulation of T-cell activation, and proliferation. Furthermore, the modified T cells exhibited robust cytotoxicity in an antigen-dependent manner, as confirmed by LDH assay results. Multiplex immunoassays, including LEGENDplex™, additionally demonstrated elevated production of cytotoxicity-associated cytokines driven by granzymes and soluble Fas ligand (sFasL). Their findings underscore the specific targeting potential of engineered TCR T-cells against NYESO1-positive tumors. Further comprehensive in vivo investigations are essential to thoroughly validate these results and effectively harness the intrinsic potential of genetically engineered T-cells for combating cancer. This study is comprehensive and well-presented and deserved to be accepted.

Author Response

Dear reviewer 2, We extend our sincere gratitude for your dedicated time and meticulous effort in reviewing our manuscript. Your valuable feedback has been both motivating and enlightening. We are pleased to note your encouraging comments on the manuscript.

Round 2

Reviewer 1 Report

No further comments

Moderate language editing